# Understanding healing: A comparative analysis in chronic diseases with leprosy—A scoping review

Joydeepa Darlong[1]*, Joy Kim[2], Subhojit Goswami[1], Chhavi Tyagi[1], Govindasamy Karthikeyan[1], Mythily Vandana S. Charles[3], Aashish Masih[1], Rama V. Baru[4]

1 The Leprosy Mission Trust India, New Delhi, India, 2 Effect hope (The Leprosy Mission, Canada), Markham, Ontario, Canada, 3 College of Nursing, Christian Medical College, Vellore, Tamil Nadu, India, 4 Jindal School of public health and human development, O.P Jindal Global University, Sonipat, Haryana, India

* joydeepa.darlong@leprosymission.in

## Abstract

### Background

Healing in leprosy has long been synonymous with bacteriological cure, often overlooking persistent disability, stigma, and psychosocial consequences. Insights from other chronic diseases may inform a broader understanding of healing. recovery.

### Objectives

To map how healing is defined and experienced in leprosy, tuberculosis, HIV/AIDS, diabetes mellitus, and schizophrenia, and to identify conceptual and practical lessons relevant for post-cure leprosy care.

### Methods

A scoping review was conducted following the Arksey and O'Malley framework and reported in accordance with PRISMA-ScR guidelines. PubMed and PsycINFO were searched for qualitative studies (January 2012–December 2022) in English language from low- and middle-income countries. Eligible studies explored definitions, determinants, or models of healing in the five conditions. Data were charted and thematically synthesized across physical, psychological, socioeconomic, socio-relational, and spiritual domains following the Joanna Briggs approach.

### Results

Eighty-five studies met inclusion criteria (leprosy = 20, TB = 9, diabetes = 8, HIV = 36, schizophrenia = 12). Healing was most often defined as adaptation, resilience, or reintegration rather than cure. Across diseases, five interrelated dimensions—physical, psychological, socioeconomic, socio-relational, and spiritual—shaped recovery.

**Data availability statement:** All data underlying the findings of this study are fully available within the paper and its Supporting information files. The review did not generate or analyse any primary data; all data were extracted from published studies identified through systematic searches of PubMed and PsycINFO. Detailed characteristics of included studies, search strategies, and synthesis tables are provided in the Supporting information.

**Funding:** This research received funding from the leprosy research Initiative Grant number FP23/100019 to JD. The funders had no role in study design, data collection and analysis, decision to publish, or preparation of the manuscript.

**Competing interests:** The authors have declared that no competing interests exist.

Compared to other chronic conditions, leprosy literature remained largely biomedical, with limited exploration of psychosocial and spiritual healing.

## Conclusions

Achieving zero leprosy requires person-centred care that embraces multidimensional healing. The proposed 5D Healing Framework offers a roadmap for integrating psychosocial, economic, and spiritual dimensions into post-cure leprosy services.

## Author summary

Healing from leprosy has been defined by bacteriological cure, yet many people continue to live with the effects of disability, stigma, and social exclusion long after treatment ends. This scoping review explored how healing is conceptualized across five chronic conditions—leprosy, tuberculosis, HIV/AIDS, diabetes mellitus, and schizophrenia—to draw lessons that can guide holistic leprosy care. By analysing 85 studies from low- and middle-income countries, we found that recovery is not limited to physical health but includes psychological, social, economic, and spiritual well-being. Other chronic diseases have progressively integrated counselling, peer support, and livelihood interventions into their care models, while leprosy programs remain largely biomedical. We propose a 5D Healing Framework—covering physical, psychological, socio-relational, socio-economic, and spiritual dimensions—to help design comprehensive, person-centred post-cure leprosy services. This approach emphasizes that achieving "zero leprosy" must include restoring dignity, belonging, and purpose for persons affected, not merely eliminating infection.

## Introduction

### Global context of leprosy: burden, disability, stigma, and gaps after MDT

Leprosy is a complex disease affecting populations in low- and middle-income countries across Asia, Africa, and Latin America [1]. It is associated with physical disability and poor psychological outcomes [2] and remains a major global health problem in the developing world [3]. While multidrug therapy (MDT) effectively cures the infection, statistics often fail to capture the suffering, disability, and dysfunction that remain after treatment completion [4]. The disease damages peripheral nerves [5], leaving many with sensory loss and muscle weakness in the hands and feet, predisposing them to injuries from daily activities. These presentations progress into chronic ulcers, loss of digits, and disfigurement—visible signs of leprosy that reinforce stigma. Studies confirm that persons affected remain at risk of developing disabilities even after being released from treatment (RFT) [4,5].

Beyond physical deformities, immune-mediated complications persist. Type 1 reactions with neuropathy may occur years after MDT completion, while type 2

reactions can be chronic, lasting a median of five years and sometimes delayed up to 16 years [6,7]. Since onset often occurs in adolescence or early adulthood, the accumulation of disability over a lifetime is substantial. Individuals continue to face stigma, poor mental health, reduced quality of life, limited social participation, and economic challenges including loss of employment [8,9].

### Rationale: why healing beyond cure matters

Despite these long-term consequences, individuals are declared 'cured' once MDT is completed, in line with the current biomedical definition of cure. However, for them, cure extends beyond bacteriological clearance to encompass the ability to return to pre disease conditions, daily activities, work, family, and social life. Current treatment pathways do not have mechanisms to address these, resulting in many who do not feel fully healed and are left stranded in their journey toward restoration [10,11] Recent studies emphasize the importance of post-cure care and have begun developing holistic care packages to support this broader concept of healing [12,13].

### Justification: lessons from chronic communicable and non-communicable diseases

Healing in leprosy has not been systematically studied however other chronic diseases like schizophrenia, diabetes, TB and HIV/AIDs may provide useful insights and parallels. Schizophrenia though dissimilar in etiology, shares with leprosy the chronic suffering, stigmatized socially that demands to look beyond the concept of cure and what it entails to manage and cope with the disease [14,15]. Similarly, Diabetes mellitus entails lifelong management of complications and stigma, including difficulties in relationships and employment that resonates with leprosy [16]. Communicable diseases like Tuberculosis and HIV/AIDS share features of stigma and long-term effects [17,18].

There is a recognition of life beyond treatment for both conditions wherein person-centred care models that have been developed by the WHO have shown the positive impact for TB survivors [19], while people living with HIV have played an important role in highlighting biases of the community and health services. Similarly, WHO guidance for HIV emphasizes differentiated, person-centred service delivery models that integrate psychosocial support and community engagement [19], while recovery-oriented approaches in mental health care promoted through WHO's mhGAP highlight social inclusion and functional recovery [20,21]. WHO frameworks on disability-inclusive development further reinforce the need to address long-term functioning, participation, and dignity beyond disease-specific treatment [22]. The voices of Persons living with HIV/AIDs have been vocal and have played an important role in reshaping attitudes of the public and within the care providers in the health services [23].Despite a broad understanding of leprosy and its management, there remains a gap in literature on what healing means for a person affected by the disease beyond bacteriological cure. While treatment completion and release from therapy (RFT) mark clinical milestones, they do not necessarily signify the end of suffering or restoration of well-being. This study, therefore, seeks to explore the broader dimensions of healing in leprosy and highlight the importance of comprehensive post-RFT care that addresses the continuing needs of persons affected.

### Aim and objectives of the review

This scoping review aimed to explore what healing means in the above-mentioned chronic diseases. In addition, the review sought to identify existing programs, service delivery approaches, and policy-relevant interventions addressing chronic suffering, stigma, and long-term recovery in low- and middle-income countries that could inform post-cure leprosym care. It also examined existing literature on leprosy to identify current understanding and gaps in conceptualizing healing. The lessons drawn will help inform the design of comprehensive care packages that support persons affected by leprosy in achieving healing beyond cure from the disease.

## Methods

### Framework and reporting

This scoping review followed the methodological framework proposed by Arksey and O'Malley (2005), further refined by the Joanna Briggs Institute (JBI), and is reported according to the PRISMA-ScR 2020 checklist [24]. The protocol was not registered due to its exploratory scoping nature.

Operationally, the focus was on mapping how healing is conceptualized across selected chronic diseases and on identifying documented models, programs, and approaches supporting multidimensional healing that may inform healing and recovery in leprosy care.

### Eligibility criteria

Inclusion: Peer-reviewed qualitative studies exploring the meaning, experience, or process of healing or recovery in adults with one of the target diseases, published between January 2012 and December 2022, in English, and from low- and lower-middle-income countries.

Exclusion: Case reports, commentaries, editorials, paediatric studies, and studies from high-income settings. Paediatric studies were excluded because this review focused on adult conceptualizations of healing related to identity reconstruction, livelihood, social participation, and meaning-making following chronic illness. Healing processes in children are mediated through caregivers, schooling systems, and developmental stages, and therefore require a distinct analytical framework beyond the scope of this review.

### Information sources and search strategy

Two databases—PubMed and PsycINFO—were searched to capture both biomedical and psychosocial perspectives. Search terms combined MeSH and free-text terms for *healing* and related constructs ("recovery," "aftercare," "well-being," "social adjustment," etc.) with disease-specific terms. Boolean operators "AND" and "OR" were used.

### Selection of sources of evidence

All retrieved citations were imported into Rayyan for de-duplication and screening. Two reviewers independently screened titles and abstracts for relevance. Full texts were then assessed against eligibility criteria. Disagreements were resolved by consensus. The study selection process is presented in a PRISMA-ScR flow diagram (Fig 1).

### Data charting process

A standardized data-charting template in Microsoft Excel captured the following information:

- Author(s), year, and country
- Study design and setting
- Disease focus
- Definition or description of healing
- Descriptions of programs, interventions, service delivery models, or policy-relevant approaches related to healing and recovery
- Key findings and recommendations

The template was pilot-tested and refined. Data extraction was conducted by one reviewer and verified by a second reviewer.

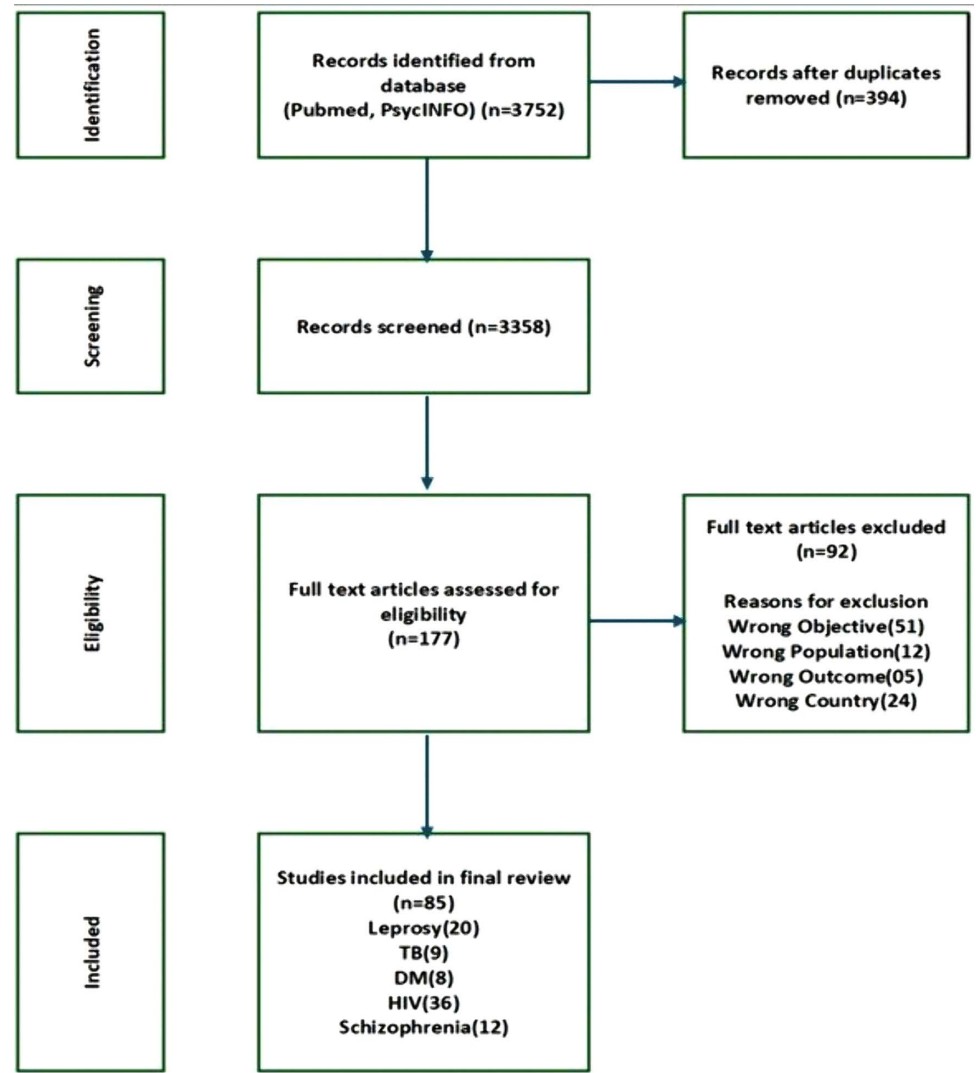

**Fig 1. PRISMA-ScR 2020 diagram of included studies.**

## Synthesis of results

Consistent with scoping review methodology, no quality appraisal was undertaken. Extracted data were summarized descriptively, and a narrative thematic synthesis identified recurring domains of healing (physical, psychological, socio-economic, socio-relational, and spiritual). Cross-cutting insights were compared across diseases to develop an integrated conceptual framework.

## Ethics

As this review used publicly available data, ethical approval was not required.

## Results

### Brief overview of included studies

In total, 3752 records were identified from two databases (PubMed and PsycINFO). After removing 394 duplicates, 3358 titles and abstracts were screened, and 3172 were excluded as irrelevant. Full texts of 177 articles were assessed for eligibility; 92 were excluded for reasons detailed in the figure. A total of 85 studies were included in the review (leprosy = 20, tuberculosis = 9, diabetes = 8, HIV = 36, schizophrenia = 12).

Table 1 summarizes the distribution of the 85 included studies across five chronic conditions and highlights regional representation and dominant methodological approaches. Detailed information on all 85 studies is presented in S1 Table.

### Lessons from chronic diseases

**Tuberculosis.** In tuberculosis, nine studies were identified, primarily from Asia, including three from India, with additional contributions from Ghana. While no one explicitly defined *healing* or *recovery*, the literature described a broad range of experiences shaping wellbeing during and after treatment. Physical aspects included early reporting of symptoms, difficulties with treatment side effects such as hearing loss, appetite changes and reduced mobility, and challenges with adherence and dietary restrictions. Socially, stigma emerged as a major theme, contributing to depression, loss of self-worth, and social withdrawal, while family support, workplace acceptance, and positive interactions with healthcare providers were described as protective. Psychological distress was commonly reported, often continuing beyond cure, though encouragement and reassurance from others were highlighted as important for adherence and recovery. Several studies also pointed to limited knowledge and misperceptions—such as reluctance to use the word "TB," doubts about curability, and reliance on herbal or spiritual remedies—while counselling and improved patient–provider communication were seen as critical for overcoming these barriers. Economic consequences were substantial, with income loss, debt, and asset pledging commonly reported, and socioeconomic status shown to correlate with quality of life. Finally, spiritual resources were invoked in coping, with reliance on God's power described as supporting individuals through the uncertainties of illness and treatment. Support for these needs was primarily accessed through public TB programs, supplemented by NGO-led counselling initiatives and, in some settings, government nutritional and cash support schemes.

**Diabetes mellitus.** From 316 identified records, eight qualitative studies were included, most of which focused on type 2 diabetes, with one involving participants with type 1 diabetes. These studies explored lived experiences through interviews and focus groups, examining coping strategies, cultural influences, and social support in self-management. Healing was not explicitly defined, but was described implicitly through practices that enabled coping, adaptation, and resilience.

Self-management emerged as a central theme, though it was often shaped by cultural and contextual factors. In some settings, patients initially did not view self-management as their own responsibility, while others emphasized the

**Table 1. Overview of included studies by disease, region, and study design.**

| Disease | Number of Studies (n = 85) | Common Regions Represented | Predominant Study Design | Key Focus Areas |
|---|---|---|---|---|
| **Leprosy** | 20 | India, Brazil, Ethiopia, Nepal | Qualitative | Healing, stigma, self-care, quality of life |
| **Tuberculosis (TB)** | 9 | India, Ghana, Bangladesh | Qualitative/ Mixed-methods | Stigma, treatment adherence, psychosocial wellbeing |
| **Diabetes Mellitus** | 8 | Ghana, India, Indonesia | Qualitative | Self-management, resilience, cultural coping |
| **HIV/ AIDS** | 36 | Sub-Saharan Africa, India, Thailand | Qualitative | Adaptation, stigma, peer support, spiritual coping |
| **Schizophrenia** | 12 | India, Turkey, Multi-country | Qualitative/ Mixed-methods | Recovery, social inclusion, spirituality |

importance of culturally tailored education and local-language training to strengthen disease control. Religious faith was another prominent coping mechanism. Prayer, fasting, and acceptance of diabetes as part of "God's plan" were seen not only as sources of strength and resilience, but also as pathways to healing and even glycaemic control [25].

Social support played a critical role in navigating the disease. Peer groups reduced isolation and fostered shared learning, while families provided emotional, financial, and practical assistance. Yet stigma and reluctance to disclose diabetes limited open discussion, contributing to hidden burdens. Cultural practices, financial constraints, and food insecurity further complicated management, with some patients turning to traditional remedies when biomedical options were inaccessible.

Across studies, the psychological impact of diabetes was clear, with participants reporting stress, frustration, and depression linked to economic and social pressures. Healing in this context was understood not as a cure, but as the ongoing ability to adapt, endure, and find meaning within the realities of chronic illness.

**Schizophrenia.** Twelve studies were included, employing qualitative, mixed-method, and scoping designs with sample sizes ranging from 9 to 282 participants. Healing was framed less as cure and more as recovery, described as a multidimensional and personal process. Key elements included adherence to or choice around medication, developing resilience, maintaining hope, and achieving psychological stability, Support systems particularly from family, community-based rehabilitation programs, and peer groups—was central to fostering belonging and reducing stigma. Employment and livelihood opportunities enhanced self-esteem and integration, while poverty and social exclusion were barriers. Spirituality and prayer were described as sources of finding meaning in life, coping, and comfort. Overall, healing in schizophrenia was defined as a continuum of recovery, combining clinical stability, psychological integration, spiritual strength, and social inclusion.

**HIV.** Thirty-five qualitative studies were included, most from Africa and Asia. Healing in HIV was not defined as cure but as a process of acceptance, adaptation, and sustaining quality of life. Clinically, barriers and facilitators to anti-retroviral therapy (ART) adherence, including provider relationships, mobility, and knowledge were significant. Socioeconomic factors, particularly family and peer support, economic stability, and participation in community networks, strongly influenced healing, while poverty and stigma hindered it. Mental health challenges such as shame, isolation, and fear of disclosure were recurrent, with acceptance of HIV status, hope, and purpose identified as key turning points. Quality of life was defined as living a "normal life," being socially accepted, and engaging in family and community roles. Spirituality was ambivalent: traditional beliefs could obstruct treatment, but faith and prayer provided coping, meaning, and motivation when combined with strong social support. Overall, healing in HIV was understood as acceptance, resilience, and integration into social and spiritual life while adhering to long-term treatment and support was commonly accessed through public ART programs, community-based peer networks of people living with HIV, NGO-led counselling services, and faith-based organizations

**Leprosy.** Thirty-eight studies, largely qualitative, examined healing among people affected by leprosy in low- and middle-income countries. Healing encompassed physical, psychological, and social recovery. Self-care education and peer support improved quality of life and reduced disability risks, but misconceptions, high turnover of medical professionals, and limited counselling skills often left patients insecure. Stigma and discrimination resulted in shame, anxiety, and depression, while peer bonding and counselling enabled resilience. Socioeconomic challenges—especially unemployment, financial insecurity, and marital rejection of women—undermined self-esteem, though microcredit, family support, and community acceptance promoted reintegration.

Interventions highlighted in the literature included early detection, health education, counselling, self-care groups, and community sensitization. Healing was thus framed as restoring dignity and reducing stigma but remained narrowly tied to physical cure and functional recovery.

In contrast, the understanding of healing in tuberculosis, diabetes, HIV, and schizophrenia has progressively expanded to integrate psychological, social, and existential dimensions. Compared to these, leprosy literature remains largely

anchored to biomedical outcomes, with limited attention to long-term psychological, socio-relational, and spiritual recovery. This support was largely delivered through vertical leprosy programs and NGO-led services, with limited integration into mainstream health systems, social protection schemes, or mental health services.

Table 2 illustrates these contrasts and exposes the gaps of leprosy recovery and how the proposed five-pillar framework addresses them.

## Discussion

This scoping review mapped and synthesized evidence on how healing is understood across five chronic conditions—leprosy, tuberculosis, diabetes, HIV, and schizophrenia. The findings highlight that healing is conceptualized as a multi-dimensional process encompassing physical, psychological, socioeconomic, socio-relational, and spiritual domains. By comparing across diseases, this review identifies lessons that can inform holistic post-cure leprosy care.

We propose describing this as the **5D Healing Framework** (Fig 2), a biological-psychological-social–economic–spiritual model.

**Physical domain.** In all chronic conditions reviewed, the physical domain remains foundational to recovery, yet the emphasis and strategies differ considerably. In tuberculosis, healing is strongly anchored in achieving microbiological cure through strict treatment adherence and consistently highlight that physician–patient communication plays a critical role in adherence and long-term health outcomes [26]. In HIV/AIDS, biomedical management is central, but layered with self-care routines such as ART adherence, symptom monitoring, and nutritional support. In diabetes, the physical aspect of healing extends beyond glycemic control to include the management of long-term complications—foot care, retinopathy screening, and lifestyle adjustments—requiring patient education and continuous self-management. In schizophrenia, while the biomedical model emphasizes medication adherence, physical healing is increasingly linked to the management of medication side effects and promotion of healthy routines such as diet, sleep, and exercise, which support overall recovery.

By contrast, in leprosy, the biomedical focus has historically been narrower, centered on multidrug therapy (MDT) completion and prevention of disability. However,physical limitations persist well beyond treatment completion, with residual nerve damage and impairments impacting daily life [27,28]. Studies in Ethiopia further highlight that integrated

**Table 2. Gaps in the current framing of leprosy recovery.**

| Chronic Disease | Healing Dimension Highlighted | Gap in Leprosy | Addressed in Five-Pillar Framework |
|---|---|---|---|
| **Tuberculosis** | Healing framed around dignity, livelihood, and social reintegration as markers of recovery | Economic security, dignity, and livelihood are rarely included in definitions of healing in leprosy | **Socioeconomic healing** – restoring dignity and livelihood through reintegration and sustainable livelihoods |
| **Diabetes** | Sustained recovery linked to family involvement and peer support in long-term self-care | Role of family and peer networks in enabling recovery is seldom acknowledged in leprosy | **Socio-relational healing** – mobilising family, peers, and community support for ongoing self-care |
| **HIV** | Recovery understood through solidarity, activism, and collective resilience | Healing in leprosy is rarely connected to solidarity, peer advocacy, or collective empowerment - Although it has started happening in some districts, but the intervention is largely dependent on NGOs | **Socio-relational healing** – strengthening peer groups, advocacy, and collective resilience |
| **Schizophrenia** | Psychological recovery emphasises hope, identity, agency, and meaning beyond symptoms | Leprosy recovery discourse remains limited to biomedical outcomes (cure, impairment prevention), with little focus on psychological dimensions | **Psychological healing** – fostering hope, agency, identity reconstruction, and resilience |
| **All diseases (crosscutting)** | Existential and spiritual dimensions often shape how recovery is experienced | Spiritual meaning-making, acceptance, and faith remain largely silent in leprosy discourse | **Spiritual healing** – integrating meaning-making, purpose, and spiritual well-being |
| **Leprosy (current framing)** | Cure, ulcer healing, and nerve function improvement | Recovery narrowly defined in biomedical terms | **Physical healing** – important but incomplete without the other four dimensions |

## The 5 Dimensions Healing Framework

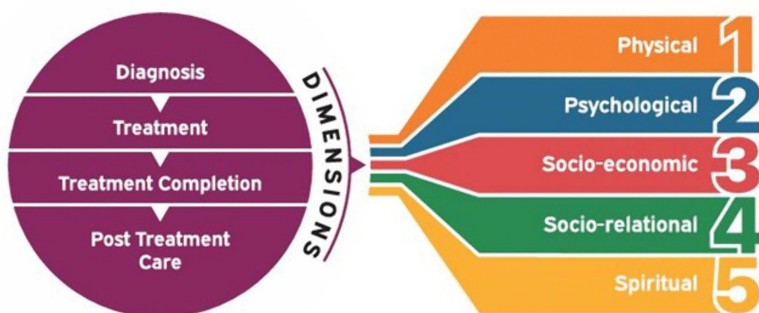

**Fig 2. The 5D Healing Framework illustrates five interrelated dimensions of healing — physical, psychological, socio-relational, socioeconomic, and spiritual — derived from cross-disease synthesis.**

approaches to limb care—such as those developed for lymphatic filariasis and podoconiosis —could be beneficial for leprosy as well, offering shared strategies for wound care, footwear provision, and physiotherapy [12]. In practice, physical care for leprosy-related complications in low- and middle-income countries is accessed primarily through public health facilities for MDT and reactions, and through NGO-supported referral centres for disability care, ulcer management, and physiotherapy. Community-based rehabilitation programs and integrated limb-care initiatives, often supported by non-governmental organizations, play a crucial role in bridging gaps in long-term physical support, particularly after release from treatment [29,30].

A key transferable lesson from TB, HIV, and diabetes is the emphasis on patient engagement in long-term physical self-care. For leprosy, this implies moving beyond clinic-based disability prevention to models that empower patients in self-care of neuropathic feet, wound management, and prevention of secondary complications. From schizophrenia and other chronic conditions, another relevant lesson is the integration of healthy lifestyle support—dietary advice, physical activity, and sleep hygiene—which are rarely addressed in leprosy but could reduce vulnerability to complications, particularly in patients on long-term corticosteroids.

Taken together, while the physical domain remains central across chronic illnesses, the framing differs. Leprosy can benefit from adopting a more holistic physical care model—one that combines MDT completion with structured, patient-led self-care, integrated chronic complication management of reactions and impairment, lifestyle support and communication strategies that build trust and adherence.

**Psychological dimension.** The psychological challenges of chronic diseases often extend far beyond the clinical course, shaping identity, relationships, and opportunities for social participation. Lessons from other long-term conditions illustrate how structured psychological support can improve adherence, recovery, and overall wellbeing.

In case of HIV, integrated counselling has become a cornerstone of care. Trained counsellors provide health education at diagnosis, during treatment initiation, and at points of crisis, ensuring patients are supported through the uncertainties and stigma of a life-altering diagnosis. Peer-support networks—often described as "treatment buddies"—play an equally important role by normalizing the illness experience, sharing coping strategies, and offering sustained encouragement for adherence.

In tuberculosis, psychosocial interventions have been crucial in addressing depression and the social isolation caused by prolonged treatment. Counselling and psychosocial support programs not only improve adherence but also reduce loss-to-follow-up. These lessons demonstrate that emotional support and treatment completion are deeply intertwined.

Schizophrenia care emphasizes recovery-oriented approaches where psychosocial interventions, including peer support groups [31], vocational rehabilitation [32], and psychoeducation for families [33], counterbalance the disabling impact of persistent symptoms. Community-based mental health teams reinforce the idea that illness management must extend beyond pharmacological control to restoring social functioning and dignity.

Diabetes programs highlight another transferable lesson: By focusing on collaborative goal-setting and empowering individuals, such approaches help patients regain agency in managing a demanding, long-term condition.

For leprosy, where stigma and social exclusion remain persistent barriers, these experiences offer clear directions for adaptation. Structured counselling at key stages—diagnosis, treatment initiation, and during episodes of reaction—could normalize the disease experience and pre-empt psychological distress. Such support, where available, is most often delivered through NGO-led counselling services, peer-support groups, and community outreach workers, rather than being embedded within routine public leprosy services [34–36]. Access remains uneven, with mental health care frequently dependent on project-based initiatives rather than integrated health system provision [37,38].

Peer-support groups modelled on HIV networks may provide safe spaces for shared learning [37–39] and reduce isolation, while involving family members through psychoeducation could help rebuild trust and support systems strained by stigma. Integrating mental health screening into routine leprosy services, alongside basic counselling skills for frontline workers, could ensure that emotional wellbeing is treated as integral to clinical care rather than an afterthought. Ultimately, a psychosocial care package for leprosy must aim not only to reduce distress but to enable participation, restore dignity, and foster resilience within both individuals and communities.

**Socioeconomic dimension.**  Experiences from other chronic conditions show that socioeconomic stability is central to sustained health outcomes. In HIV programs, the introduction of cash transfer schemes in Sub-Saharan Africa (e.g., Kenya's Cash Transfer for Orphans and Vulnerable Children) and livelihood support programs improved adherence and reduced catastrophic costs. Tuberculosis care in India and other high-burden countries has integrated nutrition support packages [40]—such as the *Nikshay Poshan Yojana*, which provides monthly cash assistance to patients during treatment—and pilot schemes for wage compensation have demonstrated positive effects on treatment completion [41,42]. In diabetes, many countries have expanded insurance coverage for essential medications and launched workplace wellness initiatives, acknowledging the dual burden of treatment costs and productivity loss. In schizophrenia, structured vocational rehabilitation programs and supported employment models—like the Individual Placement and Support (IPS) approach—have enabled social reintegration, reduced dependency, and enhanced quality of life.

For leprosy, these lessons argue for a shift from the historically charity-based rehabilitation model to a rights-based socio-economic empowerment approach. In most endemic settings, access to such support remains mediated by non-governmental organizations, with limited systematic inclusion of persons affected by leprosy in government social protection and livelihood schemes. While microfinance initiatives, cooperative enterprises, and self-help groups are already well established in leprosy work, they have often remained NGO-driven and small-scale. The challenge now is to move beyond these stand-alone models toward systemic inclusion in mainstream livelihood and social protection frameworks. Advocacy should focus on ensuring that persons affected by leprosy are fully included in government-led schemes such as disability pensions, job reservations, and national livelihood missions. At the same time, lessons from HIV and mental health programs—particularly in workplace reintegration, social protection linkages, and scale-up of community enterprises—can inform a more sustainable and rights-based economic empowerment pathway for people affected by leprosy, especially in addressing the dual burden of stigma and disability.

**Socio-relational dimension.**  The role of relationships—within families, peer groups, and communities—is a consistent determinant of long-term outcomes across chronic conditions. In HIV care, support groups and family-centred models of care have been instrumental in reducing isolation, improving adherence, and strengthening resilience. Couples counselling programs have further demonstrated benefits in disclosure, prevention, and shared responsibility for treatment. In schizophrenia, interventions such as family psychoeducation and community reintegration programs have

reduced relapse, improved functioning, and alleviated caregiver burden by building shared understanding and support networks. Tuberculosis and diabetes programs have successfully utilized patient clubs and community adherence groups, which not only improved treatment completion but also created safe spaces for peer-to-peer learning and solidarity.

For leprosy, these experiences point to the need to invest in structured self-help and family support groups. Community-based rehabilitation (CBR) should be emphasized not only for restoring functional ability but also for strengthening social belonging and reducing stigma. Finally, peer networks—modelled after successful *People Living with HIV (PLHIV) networks*—also called Champions in leprosy (Leprosy mission's Initiative) can be powerful stigma-reduction tools, providing platforms where persons affected by leprosy visibly lead advocacy, education, and mutual support, thereby normalizing participation and challenging discrimination. These peer-led initiatives are predominantly facilitated by civil society organizations and faith-based agencies, with emerging but still limited formal recognition within national leprosy programs.

**Spiritual dimension.** Spirituality has been recognized as a profound resource in coping with chronic conditions, offering both meaning and community-based support. In HIV and tuberculosis, partnerships with faith-based organizations have proven effective for stigma reduction, treatment adherence, and community mobilization. Faith leaders, when engaged positively, can shift public discourse from condemnation to compassion, influencing large networks of believers. In schizophrenia, approaches that integrate meaning-making and recovery narratives—sometimes including the individual's spiritual beliefs—have helped patients interpret their illness within a broader existential framework, building hope and resilience.

For leprosy, these insights highlight the value of collaborating with faith leaders to actively reshape community narratives, moving away from centuries-old associations of sin or curse. Recent global health discourse, including WHO-aligned calls to integrate spiritual well-being within people-centred care, provides an opportunity to formally acknowledge and responsibly integrate these dimensions within leprosy services [43,44].Care providers should also acknowledge and integrate patients' own spirituality as coping strategies, which may include prayer, meditation, or other practices that give strength during long periods of treatment and disability, enabling individuals to alter their experience of leprosy not as social exclusion but as part of a meaningful life journey.

S2 Table summarizes findings from 85 studies analyzed in the scoping review categorized by five key dimensions of healing.

## Limitations and research gaps

This review provides evidence from HIV, TB, diabetes, and schizophrenia but differences in stigma, chronicity, and programmatic infrastructure mean that transferability cannot be assumed. Publication bias, the predominance of descriptive studies, and lack of standard outcome measures further restrict the strength of conclusions.

At the same time, clear research gaps remain. Comparative implementation research is needed to test how rights-based approaches, peer-led models, and social protection schemes can be integrated into leprosy services across different contexts. Importantly, interactions across dimensions—for example, how livelihood support influences mental health or how spiritual coping affects self-care—remain underexplored. Addressing these gaps through multidisciplinary, patient-centered research is critical to advancing holistic care in leprosy.

## Cross-cutting insights and implications for practice and research

This review highlights that holistic leprosy care must extend beyond bacteriological cure to address the broader determinants of wellbeing. Across all dimensions, three cross-cutting themes emerge: the importance of community-based and patient-centred approaches, the integration of services within existing health and social protection systems, and the centrality of stigma reduction as a unifying priority. There is a need for integrated care models that recognize individuals as whole persons, situated within families, communities, and socio-cultural contexts.

In practice, this calls for the development of integrated care packages that combine biomedical treatment with mental health support, socio-economic empowerment, peer and family support networks, and spiritual resources. Embedding such approaches within primary health care and linking persons affected by leprosy to social protection schemes could improve adherence, enhance quality of life, and reduce exclusion. Stigma reduction is best achieved when these elements are mobilized together through peer-led initiatives, community-based rehabilitation, and engagement with faith and community leaders.

For research, priorities include testing care packages in leprosy contexts, using robust designs to evaluate feasibility to embed them in public health programs, their effectiveness and lived experiences. Studies should also explore the intersections between dimensions—for instance, how livelihood opportunities influence mental health, or how spirituality sustains long-term self-care. Mixed-methods trials, evaluations and longitudinal research will be particularly valuable.

Finally, for policy, the framework points to a shift from charity-based responses toward rights-based approaches. National programs must integrate psychosocial and spiritual care [45–47] strengthen disability-inclusive development, and ensure access to social protection. Such policies would not only accelerate progress toward zero leprosy but also safeguard dignity, belonging, and resilience among persons affected.

## Conclusion

This review demonstrates that achieving zero leprosy requires care that transcends bacteriological cure, embracing mental, social, economic, and spiritual dimensions alongside the biomedical. By drawing lessons from HIV, TB, diabetes, and mental health, and adapting them thoughtfully, leprosy programs can move toward genuinely holistic, person-centered care. The proposed framework offers a roadmap for practice, research, and policy—one that prioritizes dignity, resilience, and inclusion as much as medical outcomes. Future research should validate the 5D framework through mixed-methods studies in diverse leprosy-affected settings

## Supporting information

**S1 Table. Characteristics of included studies in the scoping review.** This table presents detailed information on all 85 studies included. The table corresponds to PRISMA-ScR checklist item 15.
(DOCX)

**S2 Table. Summary of outcomes across chronic diseases included in the scoping review.** This table synthesizes findings from 85 studies, categorized by five key dimensions of healing—physical, psychological, social (relational), socioeconomic, and spiritual—across five chronic conditions: leprosy, tuberculosis, diabetes mellitus, HIV/AIDS, and schizophrenia.
(DOCX)

**S3 Table. PRISMA-ScR Checklist.** This table provides a detailed mapping of the 22-item PRISMA-ScR checklist to the corresponding sections of the manuscript.
(DOCX)

**S1 Prisma Diagram. Prisma Diagram.**
(TIF)

## Acknowledgments

The authors thank all study collaborators, institutional colleagues, and individuals affected by leprosy whose experiences and insights informed this review.

## Author contributions

**Conceptualization:** Joydeepa Darlong, Joy Kim, Govindasamy Karthikeyan, Mythily Vandana S Charles, Rama V Baru.

**Data curation:** Joydeepa Darlong, Joy Kim, Subhojit Goswami, Govindasamy Karthikeyan, Aashish Masih.

**Formal analysis:** Joydeepa Darlong, Joy Kim, Subhojit Goswami, Chhavi Tyagi, Govindasamy Karthikeyan, Mythily Vandana S Charles, Aashish Masih.

**Funding acquisition:** Joydeepa Darlong.

**Project administration:** Joydeepa Darlong, Subhojit Goswami, Aashish Masih.

**Supervision:** Joydeepa Darlong, Rama V Baru.

**Writing – original draft:** Joydeepa Darlong.

**Writing – review & editing:** Joy Kim, Subhojit Goswami, Chhavi Tyagi, Govindasamy Karthikeyan, Mythily Vandana S Charles, Aashish Masih, Rama V Baru.

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
