## [Decision Letter · Decision Letter 0]

21 Jan 2026

Understanding Healing: A comparative analysis in Chronic Diseases with leprosy — A Scoping Review

Dear Dr. Darlong,

Thank you for submitting your manuscript to PLOS Neglected Tropical Diseases. After careful consideration, we feel that it has merit but does not fully meet PLOS Neglected Tropical Diseases's publication criteria as it currently stands. Therefore, we invite you to submit a revised version of the manuscript that addresses the points raised during the review process.

* A letter that responds to each point raised by the editor and reviewer(s). You should upload this letter as a separate file labeled 'Response to Reviewers '. This file does not need to include responses to any formatting updates and technical items listed in the 'Journal Requirements' section below.

* A marked-up copy of your manuscript that highlights changes made to the original version. You should upload this as a separate file labeled 'Revised Manuscript with Track Changes '.

* An unmarked version of your revised paper without tracked changes. You should upload this as a separate file labeled 'Manuscript '.

We look forward to receiving your revised manuscript.

Kind regards,

Guest Editor

Stuart Blacksell

Section Editor

Shaden Kamhawi

co-Editor-in-Chief

Paul Brindley

co-Editor-in-Chief

**Additional Editor Comments:**

Dear Author,

A well written manuscript and overall very positive reviews from the reviewers.

Regards,

Guest Editor.

**Journal Requirements:**

At this stage, the following Authors/Authors require contributions: Joydeepa Darlong. Please ensure that the full contributions of each author are acknowledged in the "Add/Edit/Remove Authors" section of our submission form.

3) Kindly revise your competing statement in the online submission form to align with the journal's style guidelines: 'The authors declare that there are no competing interests.'

**Reviewers' comments:**

Reviewer's Responses to Questions

**Key Review Criteria Required for Acceptance?**

**Methods**

-Are the objectives of the study clearly articulated with a clear testable hypothesis stated?

-Is the study design appropriate to address the stated objectives?

-Is the population clearly described and appropriate for the hypothesis being tested?

-Is the sample size sufficient to ensure adequate power to address the hypothesis being tested?

-Were correct statistical analysis used to support conclusions?

-Are there concerns about ethical or regulatory requirements being met?

Reviewer #1: - The introduction is well structured and contextualizes the challenges faced by patients in returning to their daily activities after achieving cure for leprosy, which directly impacts their quality of life and mental health. The authors also clearly articulate the reasoning behind comparing leprosy with other diseases that, from a pathophysiological perspective (except TB), do not share many similarities but do share the experience of chronic complications, suffering, and social stigma. I consider this topic highly important in the public health context, especially in low- and middle-income countries, which bear the highest burden of leprosy. However, it is still insufficiently addressed both in the scientific literature and in society at large, which makes this work even more relevant and innovative.

- Given the strong contextualization provided earlier in the introduction, I expected the objectives section to also include the investigation of more concrete methods for managing chronic suffering and social stigma resulting from the chronic diseases mentioned, rather than focusing solely on the meaning of healing for each condition. I believe it would be valuable to expand the focus to explicitly include evidence of existing programs, interventions, and public policies implemented in low- and middle-income countries for the diseases discussed.

- The methodology is very well structured and transparent. However, considering the context established in the introduction, I think it is important to include more explicitly an objective focused on searching for programs, interventions, and public policies related to the management of chronic suffering and stigma associated with the conditions analyzed.

- I find it highly relevant that the authors used as an inclusion criterion studies conducted in low- and middle-income countries, which reflect the reality of endemic countries for leprosy. It is essential to consider guidelines adopted in countries with similar social and economic conditions so that these approaches can serve as examples or inspiration when developing strategies for managing these consequences in the specific context of leprosy.

Reviewer #2: Yes, the study design is appropriate to address the stated objectives.

The authors employed a scoping review design, guided by the Arksey and O’Malley framework, refined by the Joanna Briggs Institute (JBI) methodology, and reported in accordance with PRISMA-ScR guidelines. This design is well aligned with the exploratory nature of the stated objectives .

The primary objectives—to map how healing is defined and experienced across leprosy and selected chronic diseases, and to identify conceptual and practical lessons relevant for post-cure leprosy care—require a broad synthesis of heterogeneous qualitative evidence rather than hypothesis testing or effect estimation. A scoping review is particularly suitable for:

• Clarifying complex and poorly defined concepts such as “healing” beyond biomedical cure;

• Comparing conceptual frameworks across multiple diseases;

• Identifying gaps in existing literature and informing future research and programmatic directions.

The inclusion of qualitative studies from multiple chronic conditions, the thematic synthesis across five predefined domains (physical, psychological, socioeconomic, socio-relational, and spiritual), and the absence of formal quality appraisal are all consistent with accepted scoping review methodology and appropriate for the study’s aims .

Importantly, the design allows for cross-disease comparison and conceptual model development (the proposed 5D Healing Framework), which would not have been feasible using more restrictive designs such as systematic reviews focused on narrowly defined outcomes.

In conclusion, the scoping review design is methodologically sound and well suited to address the stated objectives, providing an appropriate framework for mapping concepts, synthesizing qualitative evidence, and generating a multidimensional, practice-oriented framework for post-cure leprosy care.

**Results**

-Does the analysis presented match the analysis plan?

-Are the results clearly and completely presented?

-Are the figures (Tables, Images) of sufficient quality for clarity?

Reviewer #1: - The literature review conducted to define “healing” in the context of the chronic diseases considered in this study is extremely important for synthesizing a unified conceptual understanding and for identifying similarities and differences among the conditions. In the results section, it becomes clear that this process enabled the identification of important gaps in the concept of healing for leprosy. I consider this essential for determining the key aspects that should guide the development of strategies and recommendations for managing the main consequences of the disease. However, I felt the discussion lacked a more explicit presentation of plans or guidelines already adopted for the other diseases, which could serve as concrete examples or inspiration for the leprosy context.

- I found it very interesting and valuable that in the discussion the authors proposed a structural model, outlining the five key areas to be considered for the effective management of the consequences of leprosy, which can persist during treatment and even after biomedical cure.

- I also appreciated the structure adopted in the discussion, addressing how each of the five areas of the proposed model is handled in the context of each disease mentioned. However, I think it would be important to discuss where and how patients in low- and middle-income countries actually obtain support to meet each of these needs, as the current text feels somewhat abstract. For example, regarding the physical domain, where do patients access the necessary care for managing disease-related consequences? In public hospitals? Through nongovernmental organizations? Via governmental support programs? And do patients truly have access to these services in real-world settings?

Reviewer #2: The results are clearly, logically, and comprehensively presented.

The results section follows a coherent and transparent structure that is consistent with PRISMA-ScR reporting standards. The authors clearly describe the study selection process, including numbers at each stage, supported by a PRISMA flow diagram. The characteristics of the included studies are systematically summarized, and results are then organized thematically by disease group and by domains of healing .

Key strengths in result presentation include:

• Clear reporting of the number of included studies (n = 85) and their distribution across diseases and regions.

• A structured narrative synthesis that explicitly addresses the review objectives.

• Consistent comparison across diseases, which allows the reader to understand similarities, contrasts, and gaps—particularly the relative biomedical focus in leprosy literature.

• Logical progression from descriptive mapping to higher-level synthesis, culminating in the proposed 5D Healing Framework.

Given the exploratory nature of a scoping review, the level of detail is appropriate and sufficient. The results adequately support the study’s conclusions and proposed framework without overstating findings.

The figures and tables are of good quality and effectively enhance clarity and comprehension.

The manuscript includes well-designed and relevant visual elements:

• PRISMA-ScR flow diagram (Figure 1) clearly illustrates the study selection process and enhances transparency.

• Tables 1 and 2 concisely summarize study characteristics and cross-disease gaps in conceptualizing healing, facilitating rapid comparison.

• Figure 2 (The 5D Healing Framework) is conceptually clear, visually coherent, and well integrated with the narrative synthesis, effectively translating complex qualitative findings into an accessible model .

Overall, the tables and figures are legible, appropriately labeled, and directly linked to the text. They add explanatory value rather than redundancy and are of sufficient quality to support understanding by both academic and policy-oriented audiences.

In summary, both the results and the accompanying figures/tables are clearly and completely presented, methodologically appropriate, and well aligned with the study objectives and design.

**Conclusions**

-Are the conclusions supported by the data presented?

-Are the limitations of analysis clearly described?

-Do the authors discuss how these data can be helpful to advance our understanding of the topic under study?

-Is public health relevance addressed?

Reviewer #1: The results obtained from this scoping review are consistent with the objectives. The information synthesized regarding the concept of healing across other chronic diseases is highly relevant. Moreover, the idea of comparing these concepts across conditions to generate a concrete model of the key areas required for managing the consequences of leprosy is, in my view, innovative. Although I highlighted some points for improvement, I believe the insights presented in this work will be valuable for guiding the development of concrete action plans to improve the quality of life of people affected by leprosy.

Reviewer #2: The conclusions are well supported by the data presented.

The conclusions directly derive from the mapped evidence and thematic synthesis of the 85 included qualitative studies. The finding that healing across chronic diseases is predominantly conceptualized as a multidimensional process—extending beyond biomedical cure to include psychological, socio-relational, socioeconomic, and spiritual dimensions—is consistently demonstrated across disease-specific result sections. The conclusion that leprosy literature remains comparatively biomedical is explicitly supported by cross-disease comparisons and summarized in Table 2 and the narrative synthesis .

The proposed 5D Healing Framework is not speculative; it is grounded in recurrent themes identified across multiple disease contexts and systematically justified through the results. Thus, the conclusions are proportionate to the evidence and do not overreach the data.

Limitations are clearly and appropriately acknowledged.

The authors explicitly discuss several key limitations, including:

• The descriptive and exploratory nature of scoping reviews, which precludes causal inference;

• The predominance of qualitative studies and lack of standardized outcome measures;

• Potential publication bias;

• Contextual variability across diseases and settings, limiting direct transferability of findings to leprosy programs;

• The absence of formal quality appraisal, which is consistent with scoping review methodology but nonetheless acknowledged as a constraint .

These limitations are transparently presented and framed as areas for future research rather than undermining the validity of the review.

The manuscript makes a clear and meaningful contribution to advancing conceptual understanding.

The authors explicitly articulate how synthesizing evidence from other chronic diseases advances understanding of “healing beyond cure” in leprosy. By situating leprosy within a broader chronic disease paradigm, the study reframes healing as a person-centered, longitudinal process, rather than a discrete clinical endpoint. The discussion clearly identifies conceptual gaps in leprosy care and demonstrates how insights from HIV, TB, diabetes, and mental health can inform more holistic post-cure approaches .

Furthermore, the identification of research gaps and proposed directions for mixed-methods and implementation research illustrates how the findings can guide future scholarship and program design.

Yes, Public health relevance is strongly and explicitly addressed.

Public health relevance is a central strength of the manuscript. The findings are directly linked to:

• The global goal of “zero leprosy”, emphasizing that elimination cannot be achieved through bacteriological cure alone;

• The need for integrated, person-centered care models embedded within primary health care and social protection systems;

• Stigma reduction, disability-inclusive development, and rights-based approaches as public health priorities.

The authors clearly articulate implications for policy, health systems, and programmatic design, making the review highly relevant for public health practitioners, policymakers, and global NTD stakeholders .

Overall assessment:

The conclusions are robustly supported by the data, limitations are transparently discussed, the contribution to conceptual and practical understanding is clear, and the manuscript demonstrates strong and explicit public health relevance.

**Editorial and Data Presentation Modifications?**

Reviewer #1: - The “objectives” subsection within the methodology is somewhat redundant, since the study’s objectives were already presented in a similar section in the introduction.

- I was unsure about the exclusion of pediatric studies. The diseases analyzed also affect children, and even though the concept of “healing” may be inherently adult-oriented, this does not necessarily make pediatric contexts incompatible. Why exactly were these studies excluded? I believe the rationale behind this criterion should be better explained in the methodology.

- In Figure 1, the “records after duplicates removed” box could be moved to the side (with a horizontal arrow), placed between the first and third boxes. This would make the flow more intuitive and avoid the initial impression that the number of studies decreases and then increases again throughout the diagram, which may cause confusion at first glance.

- In the introduction, the authors briefly mention WHO programs for TB, but I did not see further references to other WHO programs or guidelines for the other diseases discussed. I believe this should be incorporated into the text, as it is important to support several of the points raised, given that the WHO is a global reference authority.

- Throughout the discussion, in the subsections addressing each of the model’s components, I repeatedly felt that citations were missing to support several statements. The authors should revisit this and include the appropriate references.

Reviewer #2: (No Response)

**Summary and General Comments**

Reviewer #1: (No Response)

Reviewer #2: I would like to further emphasize an important strength of this manuscript that merits explicit recognition:

The transversal approach adopted by the authors and the proposed paradigm shift in the definition of “cure” are particularly significant. By moving beyond a narrowly biomedical understanding of cure toward a multidimensional concept of healing, the study addresses a long-standing gap that affects not only clinical practice but also social, cultural, and institutional responses to leprosy.

This paradigm shift is highly relevant from the medical perspective, as it highlights that bacteriological cure does not equate to restoration of health, functioning, or quality of life. Equally important, it is relevant from social, religious, and cultural perspectives, as stigma, moral interpretations of disease, spiritual meaning-making, and community belonging continue to shape the lived experience of persons affected by leprosy long after treatment completion.

By integrating medical, psychosocial, socioeconomic, socio-relational, and spiritual dimensions in a transversal manner, the manuscript contributes to a more ethically grounded, person-centred, and rights-based understanding of healing. This approach aligns with contemporary thinking in global health and chronic disease management and is essential for designing post-cure interventions that are truly responsive to patients’ realities.

Overall, this transversal and paradigm-shifting perspective substantially enhances the conceptual depth, novelty, and societal relevance of the study.

This is a high-quality, conceptually strong, and policy-relevant manuscript. The strengths substantially outweigh the limitations. I consider it a valuable contribution to the field and well suited for publication in PLOS Neglected Tropical Diseases.

PLOS authors have the option to publish the peer review history of their article (what does this mean? ). If published, this will include your full peer review and any attached files.

**Do you want your identity to be public for this peer review?** For information about this choice, including consent withdrawal, please see our Privacy Policy .

Reviewer #1: **Yes:** Isabela Espasandin

Reviewer #2: No

**Figure resubmission:**
---

## [Editor Report · Decision Letter 1]

9 Feb 2026

Dear Dr Darlong,

We are pleased to inform you that your manuscript 'Understanding Healing: A comparative analysis in Chronic Diseases with leprosy — A Scoping Review' has been provisionally accepted for publication in PLOS Neglected Tropical Diseases.

Best regards,

Udhishtran Arudchelvam

Guest Editor

Stuart Blacksell

Section Editor

Shaden Kamhawi

co-Editor-in-Chief

Paul Brindley

co-Editor-in-Chief

---

## [Editor Report · Acceptance letter]

Dear Dr Darlong,

We are delighted to inform you that your manuscript, "

Understanding Healing: A comparative analysis in Chronic Diseases with leprosy — A Scoping Review," has been formally accepted for publication in PLOS Neglected Tropical Diseases.

Best regards,

Shaden Kamhawi

co-Editor-in-Chief

Paul Brindley

co-Editor-in-Chief
